# Seismic Response of Resilient Bridges with SMA-Based Rocking ECC-Reinforced Piers

**DOI:** 10.3390/ma14216500

**Published:** 2021-10-29

**Authors:** Xiaogang Li, Ke Chen, Ji Chen, Yi Li, Dong Yang

**Affiliations:** 1PowerChina Huadong Engineering Corporation Limited, Hangzhou 311122, China; li_xg2@hdec.com (X.L.); chen_j7@hdec.com (J.C.); li_y11@hdec.com (Y.L.); yang_d2@hdec.com (D.Y.); 2Research Institute of Highway, Ministry of Transport, Beijing 100088, China

**Keywords:** seismic analysis, rocking pier, shape memory alloy, ECC material, bridge engineering

## Abstract

Post-earthquake investigation shows that numerous reinforced concrete (RC) bridges were demolished because of large residual displacements. Improving the self-centering capability and hence resilience of these bridges located in earthquake-prone regions is essential. In this regard, a resilient bridge system incorporating engineered cementitious composites (ECC) reinforced piers and shape memory alloy (SMA) energy dissipation components, i.e., SMA washers, is proposed to enhance its resilience when subjected to strong earthquakes. This study commences with a detailed introduction of the resilient SMA-washer-based rocking bridge system with ECC-reinforced piers. Subsequently, a constitutive model of the ECC material is implemented into OpenSees and the constitutive model is validated by test data. The working principle and constitutive model of the SMA washers are also introduced. A series of dynamic analysis on the conventional and resilient rocking bridge systems with ECC-reinforced piers under a suite of ground motions at E1 and E2 earthquake levels are conducted. The analysis results indicate that the resilient rocking bridge system with ECC-reinforced piers has superior resilience and damage control capacities over the conventional one.

## 1. Introduction

Post-earthquake field investigations on the damaged bridges revealed that many reinforced concrete (RC) bridges, although designed conforming to the ductility design philosophy, commonly experienced overly large residual displacement which is difficult to recover. For example, more than 100 RC piers were demolished because they suffered from large permanent drift ratio (i.e., 1.5%) after the 1995 Kobe earthquake [1]. Lessons drawn from these events enlighten us that only satisfying the seismic ductility demand is not enough for engineering structures because their residual deformation after earthquake still significantly jeopardizes their normal functionality [2,3,4]. To guarantee service operation of the structures after earthquakes, resilient capacity is being paid more attention in the seismic codes of many countries (e.g., US, Japan, and New Zealand) [5]. Rocking component, as a resilient structural member, has been attracting extensive experimental and numerical studies [6,7,8,9,10]. For instance, the conventional post-tensioned (PT) rocking bridges have been studied by shake table tests recently [11,12,13]. These studies revealed that these self-centering bridge systems were capable of sustaining a large drift ratio of up to 10% but only experienced small residual drift ratio (i.e., 0.5%) with non-critical damages [14]. Subsequently, a series of innovative devices had been presented to further improve the self-centering and energy dissipation capacities of the rocking piers under extreme earthquake events [15,16,17,18]. Although the PT tendons together with various energy dissipaters can provide excellent recoverability and energy dissipation capacity to the rocking pier [19,20], the energy dissipater may be damaged and thus should be replaced after earthquakes, leading to compromised rescue efficiency. Additionally, some damage patterns such as relaxation and environmental corrosion of the PT tendons are difficult to fix. In this regard, shape memory alloy (SMA) that is characterized by super-elasticity has been recently considered for various devices (i.e., SMA tendons, bars, and springs) used in resilient bridge structures [21,22,23,24,25,26,27,28,29,30,31] as well as other types of structural systems [32,33,34,35,36,37,38,39]. In particular, a bridge system with SMA-washer based rocking pier was recently proposed to achieve self-centering functionality during earthquakes [40]. The SMA washers provided restoring force for the RC pier, which can eliminate some inherent shortcomings, such as corrosion and relaxation, induced by the PT tendon. However, the reinforcing steel embedded in the plastic hinge of the pier was still vulnerable to yield due to large bending moment during severe earthquakes. Varela and Saiidi [41] integrated SMA bars with elastomeric rubber bearing to replace the conventional plastic hinge of the RC pier. The test results indicated that, except for the bucking of the SMA bars, the RC pier experienced almost no damage even under 2.5 times the design’s seismic loading. One drawback of the SMA bars is their sensitivity to manufacture imperfections and vulnerable to brittle fracture when sustaining large strains [42,43]. Apart from the SMA material, an innovative engineered cementitious composites (ECC) material was used to replace the conventional concrete in the plastic hinge of the pier to enhance its strength and ductility in both tension and compression [44]. The metal-like strain strengthening property of the ECC material makes it an attractive alternative to the conventional concrete for earthquake resilient structures.

Inspired by the existing studies on rocking piers and the application of emerging materials such as ECC and SMA, a resilient self-centering structural bridge system with SMA-washer-based ECC-reinforced rocking piers is proposed. The SMA washers allows the pier to rock in a large rotation angle under reliable control. The ECC material in the pier can significantly enhance the resilience of the rocking piers and remarkably reduces structural cracks during earthquakes. In this paper, the configuration and self-centering rocking mechanism of the resilient bridge system with ECC reinforced pier is firstly proposed. Subsequently, the constitutive model of the ECC material is introduced and validated by an experimental study on a 1/5 scaled ECC-reinforced pier. An FE model of the prototype rocking bridge with the SMA-washer-based ECC-reinforced rocking pier is established in OpenSees [45] to study the superiority of the proposed solution in alleviating seismic damage over the conventional bridges.

## 2. Resilient Self-Centering Rocking Bridge System with ECC-Reinforced Pier

### 2.1. Configuration of the Rocking Bridge System

The configuration of the resilient bridge system with the ECC-reinforced rocking pier is presented in Figure 1. The main body of the pier and two separated pile caps (i.e., the upper pile cap and the lower pile cap) are the most critical components in the resilient bridge system. Each SMA washer set consists of a group of SMA washer springs and associated nuts, anchor rebar and shim plates. The anchor rebar was casted into the concrete of the lower pile cap and passed through the upper one via a plastic tube which was embedded in the upper pile cap. The SMA washers which were assembled in appropriate styles (e.g., in parallel, in series or in combination) passed through the anchor rebar and were tightly precompressed by the nuts. This SMA-washer based rocking solution is capable of providing stable recoverability and moderate energy dissipation capacity for the bridge system when subjected to severe earthquake events. The fixed bearings were installed on the top of the bent cap but sliding bearings are installed on the top of two abutments for the girder deformation induced by temperature variance. A comprehensive experimental study on reduced scale SMA-washer based rocking piers (with normal concrete) has been recently conducted, where the details are reported elsewhere [40]. When ECC is used for the pier, extra benefits are enabled such as enhanced tension and compression strength and ductility to sustain flexural and shear deformation during seismic excitations. In this way, the residual deformation of the resilient bridge system tends to decrease and the damage to the pier will be alleviated. For the prototype bridge, six sets of the SMA washer devices are equipped in the pile. The configuration of a typical SMA washer set is shown in Figure 1b.

### 2.2. Rocking Mechanism and Seismic Design Objectives of Resilient Bridge System

There are three seismic design objectives under different earthquake intensities (i.e., small earthquake, moderate earthquake and large earthquake). The first objective is that the interface between two pile caps is close under small earthquakes, which ensures that the rocking bridge system exhibits similar function to the conventional bridge with fixed base pier. The second one is that the maximum drift ratio of the girder is not more than 1.0% under moderate earthquakes (i.e., E1 level). Such a small drift angle would induce limited damage to the structure. The last one is that the maximum drift ratio of the girder is not more than 2.0% under large earthquakes (i.e., E2 level). When the bridge subjects to severe earthquakes, the resulting inertial force will cause the bridge to rock around the two base corners of the upper pile cap and meanwhile the compressed SMA washers together with the gravity of the bridge will provide restoring force for the bridge to return to its original state. The maximum rocking angle of the pier can be controlled by designing appropriate assembles (i.e., series, parallel or both of them) of the SMA washer set. The natural period of the resilient bridge is much larger than that of the conventional bridge, which makes it far away from the dominant periods of the earthquake ground motions. Thus, the resilient bridge system is expected to significantly reduce seismic damage.

The rocking mechanism of the pier and the seismic design objectives of the bridge can be interpreted by Figure 2. The total height of the pier and the pier cap is H. To satisfy the first seismic design objective, an appropriate prestressed force should be imposed on the SMA washer sets, as shown in Figure 2a. The uniform distribution of the resulting reacting force at the bottom of the upper pile cap is also shown in Figure 2a. The upper pile cap and the bottom pile cap always touch tightly under dead load of the superstructure (i.e., G) and servicing loads such as vehicle and temperature. When the horizontal force F_h_ continuously increases, the reacting force at one base side gradually reduces to zero, as shown in Figure 2b. If F_h_ further increases, the pier will uplift with a drift ratio of α = Δ/H, where Δ is the horizontal displacement of the pier, as shown in Figure 2c. The horizontal displacement is commonly composed of two parts, of which one part is the deformation of the pier and another part is the contribution of rocking. If the horizontal deformation of the RC pier is too large, the bottom of the pier may yield. Therefore, the ECC material is proposed in this study as an alternative to the conventional concrete material for the pier. The ultimate drift ratio α_u_ can be calculated by Δ_u_/H, as shown in Figure 2d. When the designated ultimate drift ratio (i.e., 2.0%) reaches, the SMA washer sets at one side of the pier is fully flattened. If the drift ratio of the bridge exceeds 2.0% during earthquakes, the pier will yield. The proposed resilient rocking bridge system is capable of self-locking, which is a unique property over the conventional one.

## 3. Verification of ECC Material Constitutive Models

### 3.1. Constitutive Models of ECC Material

In order to study the seismic response of the ECC-reinforced structures, a constitutive model [45,46], which is capable of considering the computational efficiency and accuracy, is essential for establishing finite element (FE) model, e.g., OpenSees model. The envelop curves of the constitutive model is composed of two parts, of which one is related to the tension state (i.e., *O-A-F-I*) and the other part is associated with the compression state (i.e., *O-J-P*), as shown in Figure 3. The envelop curve in tension is multilinear curves which are expressed as Equation (1), and the envelop curve in compression is also the multilinear curves as written in Equation (2).

Segment *A*-*F* in the tension region is a strain hardening stage, which indicates that the ECC material is more resilient than the normal concrete. The three segments on the envelope curve within the tensile region (i.e., segments *O*-*A*-*F*-*I*) are given in
(1)$$F_{t}=\left\{\begin{array}{ll} E\varepsilon & 0\leq \varepsilon \leq \varepsilon_{t0}\\ \sigma_{t0}+\left(\sigma_{tp}-\sigma_{t0}\right)\frac{\left(\varepsilon -\varepsilon_{t0}\right)}{\left(\varepsilon_{tp}-\varepsilon_{t0}\right)} & \varepsilon_{t0}\leq \varepsilon <\varepsilon_{tp}\\ \sigma_{tp}\frac{\left(\varepsilon_{tu}-\varepsilon \right)}{\left(\varepsilon_{tu}-\varepsilon_{tp}\right)} & \varepsilon_{tp}\leq \varepsilon <\varepsilon_{tu}\\ 0 & \varepsilon_{tu}\leq \varepsilon \end{array}\right.$$
where *E* is Young’s modulus. The first microcrack will occur if the strain ε is larger than the cracking strain $\varepsilon_{t0}$ at point *A*. The loading path further follows the segment *A*-*F* until the strain reaches the peak tensile strain $\varepsilon_{tp}$ at point *F*, where the stress simultaneously reaches the peak tensile stress $\sigma_{tp}$. However, a soft stiffness (i.e., segment *F*-*I*) will occur if the tensile strain exceeds $\varepsilon_{tp}$ but is smaller than the ultimate tensile strain $\varepsilon_{tu}$ at point *I*, where the corresponding stress becomes zero. It is worth noting that the loading path will continuously move forward along the positive abscissa axis if the strain is larger than $\varepsilon_{tu}$.

The envelope curve consists of multilinear curves in the compressive region (i.e., segments *O*-*J*-*P*) which can be expressed as
(2)$$F_{c}=\left\{\begin{array}{ll} E\varepsilon & \varepsilon_{tp}\leq \varepsilon <0\\ \sigma_{cp}\frac{\left(\varepsilon_{cu}-\varepsilon \right)}{\left(\varepsilon_{cu}-\varepsilon_{cp}\right)} & \varepsilon_{cp}\leq \varepsilon <\varepsilon_{cu}\\ 0 & \varepsilon_{cu}\leq \varepsilon \end{array}\right.$$
where $\sigma_{cp}$ and $\varepsilon_{cp}$ are the peak compressive stress and corresponding strain, respectively. $\varepsilon_{cu}$ is the ultimate compressive strain. It is noted that loading path will further move forward along the negative abscissa axis if the strain is smaller than $\varepsilon_{cu}$.

The loading, the unloading and the reloading rules of the ECC model in the tension region (see Figure 4a) are given by$$\;F_{t}=\left\{\begin{array}{lll} E\varepsilon & 0\leq \varepsilon_{tm}<\varepsilon_{t0} & \;(3)\\ \sigma_{tl}^{\prime }{\left(\frac{\varepsilon -\varepsilon_{tl}}{\varepsilon_{tm}^{\prime }-\varepsilon_{tl}}\right)}^{\alpha_{t}} & \varepsilon_{t0}\leq \varepsilon_{tm}<\varepsilon_{tp},\overset{}{\dot{\varepsilon }<0} & \;(4)\\ \sigma_{tl}^{\prime }+\left(\sigma_{tm}-\sigma_{tl}^{\prime }\right)\left(\frac{\varepsilon -\varepsilon_{tl}^{\prime }}{\varepsilon_{tm}^{\prime }-\varepsilon_{tl}^{\prime }}\right) & \varepsilon_{t0}\leq \varepsilon_{tm}<\varepsilon_{tp},\overset{}{\dot{\varepsilon }\geq 0} & \;(5)\\ \sigma_{tm}\left(\frac{\varepsilon -\varepsilon_{tl}}{\varepsilon_{tm}-\varepsilon_{tl}}\right) & \varepsilon_{tp}\leq \varepsilon_{tm}<\varepsilon_{tu} & \;(6) \end{array}\right.$$
where $\alpha_{t}$ is a constant that is larger than or equal to 1. It can be calibrated using the experimental data. The segment *B*-*C*-*E* is the initial unloading path expressed by the Equation (4). $\varepsilon_{tm}^{}$ is the maximum strain reached in history on the envelop curve where the unloading is triggered. $\varepsilon_{tl}^{}$ is the strain corresponding to the stress vanishing on the initial unloading path (i.e., the strain associated with the point *E* or *H*). The value $\varepsilon_{tl}^{}=\beta_{t}\cdot \varepsilon_{tm}$, where $\beta_{t}$ is a constant. Segment *C*-*D* is a typically partial reloading path, where the stress at point *C* is not zero, expressed by Equation (5), which ensures the extension of the segment *C*-*D* passing through the historically reached maximum strain point on the envelop curve (i.e., point *B*). The $\varepsilon_{tr}^{}$ and $\varepsilon_{tu}^{}$ are strains at points *C* and *D*, respectively. The unloading path starting from *D* is controlled by Equation (4). The subscript on the parameter $\varepsilon_{tm}^{\prime }$ denotes that $\varepsilon_{tm}^{\prime }$ should be set as $\varepsilon_{tm}^{}$ or $\varepsilon_{tr}^{}$ when they are used to define the initial unloading path or partially reloading path, respectively. The same specification applies to the parameters $\varepsilon_{tl}^{\prime }$ and $\varepsilon_{tu}^{\prime }$.

The unloading and reloading paths after point *F* (i.e., *F*-*G*) are a linear curve given by Equation (6). The loading, the unloading and the reloading paths of the ECC model in the compression region (see Figure 4b) are expressed as:$$\;F_{c}=\left\{\begin{array}{lll} E\varepsilon & \varepsilon_{cp}\leq \varepsilon_{cm}<0 & \;(7)\\ \sigma_{cm}^{\prime }{\left(\frac{\varepsilon -\varepsilon_{cl}}{\varepsilon_{cm}^{\prime }-\varepsilon_{cl}}\right)}^{\alpha_{c}} & \varepsilon_{cu}\leq \varepsilon_{cm}<\varepsilon_{cp},\dot{\varepsilon }\overset{}{>0} & \;(8)\\ \sigma_{cu}^{\prime }+\left(\sigma_{cm}-\sigma_{cm}^{\prime }\right)\left(\frac{\varepsilon -\varepsilon_{cl}^{\prime }}{\varepsilon_{cm}^{\prime }-\varepsilon_{cl}^{\prime }}\right) & \varepsilon_{cu}\leq \varepsilon_{cm}<\varepsilon_{cp},\dot{\varepsilon }\overset{}{\leq 0} & \;(9) \end{array}\right.$$

The segment *K*-*N*, as presented in Figure 4b, is formulated by Equation (8). $\varepsilon_{cm}^{}$ is the strain on the envelope curve where unloading is triggered (i.e., the strain at point *K*) and the $\varepsilon_{cl}^{}$ is the strain on the initial unloading path corresponding to zero stress (i.e., the strain at point *L*). The value $\varepsilon_{cl}^{}=\beta_{c}\cdot \varepsilon_{cm}$, where $\beta_{c}$ is a constant. Segment *N*-*M* is formulated by Equation (9). $\varepsilon_{cr}^{}$ and $\varepsilon_{cu}^{}$ are strains of the points *M* and *N*, respectively. The unloading path from *N* is given by Equation (8). The parameter $\varepsilon_{cm}^{\prime }$ should be set to $\varepsilon_{cm}^{}$ or $\varepsilon_{cr}^{}$ when they are employed to define the initial unloading path or partially reloading path.

### 3.2. ECC-Reinforced Column and Numerical Verification

A 1/5 scaled ECC-reinforced column [47,48], as shown in Figure 5a, is used to verify the effectiveness of the ECC constitutive model proposed in Section 3.1. To provide lateral load on the top of the cantilever column, a rigid ECC transverse beam is monolithically casted with the cantilever base. This loading configuration is chosen to promote a flexural deformation mode in the specimen and to investigate the effect of ECC material properties on the expected plastic hinge region in particular. Longitudinal steel reinforcement was bent at a 90º angle at the bottom of the transverse beam and further extended to provide sufficient anchorage. A total of four reinforcing steels with a diameter of 10 mm are arranged at the four corners of the cross section of the pier, resulting in a reinforcement ratio of 3.14%. The compressive strength of the ECC material is 80.0 MPa at a strain of 1.2%. The tensile strength is 6.0 MPa at a strain of 6.0%. The Young’s modulus of the ECC is 16,000.0 MPa. The Passion’s ratio of the ECC material is 0.15. The yielding strength of the reinforcing steel is 410.0 MPa at a strain of 0.02% and the ultimate uniaxial strength is 640.0 MPa at a strain of 14.0%. The lateral load protocol for the quasi-static test is shown in Figure 5b.

The FE model of the ECC-reinforced column is established in OpenSees (Version 2.4.1). A total of 5 displacement-based beam-column elements are used to model the ECC column. The *ECC02 material model* that has been developed and implemented into OpenSees is employed to capture the response of the column. The analysis results regarding the horizontal force against drift ratio together with the test results are displayed in Figure 6. It indicates that the numerical simulation results agree well with the test results. It confirms that the ECC constitutive model proposed in this study is sufficiently accurate for further dynamic analysis.

### 3.3. Constitutive Models of SMA Washer

The chemical composition of the SMA washer is 55.87% nickel and 44.13% titanium alloys by weight and it is supposed to exhibit super elasticity at room temperature (austenite finish temperature is 4.5 °C). The constitutive model of the SMA washer has been developed in [40], which has a flag-shaped envelop curves, including loading stage (i.e., O-A-B-E) and unloading stage (i.e., E-B-C-E-O), as shown in Figure 7, where *F*_2_ and *δ*_2_ represent the “yield” force and the corresponding deformation, respectively; *F*_3_ and *δ*_3_ are the force and deformation when the SMA washer is fully flattened. Similarly, *F*_4_, *δ*_4_, *F*_1_, and *δ*_1_ represent the characteristic forces and deformations during the reverse plateau. Once the SMA washer reaches its maximum compressive deformation, the axial stiffness increases abruptly (i.e., BE).

The loading stag on the envelop curve is composed of three segments, which are expressed as: (10)$$F=\left\{\begin{array}{ll} E_{w}\delta & 0\leq \delta <\delta_{2}\\ F_{2}+k_{1}E_{w}\left(\delta -\delta_{2}\right) & \delta_{2}\leq \delta <\varepsilon_{3}\\ F_{3}+k_{2}E_{w}\left(\delta -\delta_{3}\right) & \varepsilon_{3}\leq \delta \end{array}\right.$$where *E_w_* is the elastic stiffness of the single SMA washer; *k*_1_ and *k*_2_ are constants which can be determined from test.

The unloading stage on the envelop curve consists of four segments, which are given as follows:
(11)$$F=\left\{\begin{array}{ll} E_{w}\delta & 0\leq \delta <\delta_{1}\\ F_{4}-k_{1}E_{w}\left(\delta_{4}-\delta_{}\right) & \delta_{1}\leq \delta <\delta_{4}\\ F_{3}-E_{w}\left(\delta_{3}-\delta_{}\right) & \delta_{4}\leq \delta <\delta_{3}\\ F_{3}+k_{2}E_{w}\left(\delta_{}-\delta_{3}\right) & \delta \geq \delta_{3} \end{array}\right.$$

The uploading and reloading stiffness away from the envelop curve is *E_w_*.

The key parameters for a single SMA washer are shown in Figure 7.

The diagram of a typical SMA washer is presented in Figure 8, where the outer and inner diameters of the SMA washer are D and d, respectively. The thickness and the height of the SMA washer are h and t, respectively. These parameters of a SMA washer can be appropriately adjusted according to the seismic design objectives of rocking bridges. Once all the parameters are designated, the maximum compressive deformation and the restoring force provided by a SMA washer can be calculated.

## 4. Validation of the Numerical Simulation Method for Capturing Rocking Behavior

A 1/4 scale RC rocking pier specimen equipped with SMA washer sets [29] is employed to validate the numerical simulation method used in this study for capturing the overall rocking behavior. The configuration of the rocking pier is shown in Figure 9. The test specimen includes one bent cap, the main body of a pier, two separated pile caps, four SMA washer sets, and a tendon. The diameter of the RC pier is 0.3 m. The strength of the normal concrete was 39.0 MPa at the test day. A total of 12 longitudinal HRB400 rebars with a diameter of 16 mm were uniformly arranged along the perimeter, resulting in a reinforcement ratio of 0.3%. The average yielding strength of the longitudinal rebar was 400.4 MPa. The diameter of the transverse reinforcement was 10 mm and the space between two adjacent stirrups was 75 mm. A total of four SMA washer sets were installed on the top of the upper pile cap and each SMA washer set composed of the loading protocol is shown in Figure 10. Other information regarding this rocking pier can be find elsewhere [29].

To validate the numerical simulation method proposed in this study for capturing the overall rocking behavior, the experimental results of the rocking pier specimen are used to make comparison with the numerical analysis results. The RC pier was divided into seven displacement-based fiber beam-column elements, of which the normal concrete fiber was assigned with *Concrete02 material model* and the reinforcement fiber was assigned with *Steel02 material model*. The connection between the pier and the bent cap or the pile cap is modeled by rigid elements. The interface between upper and lower pile caps was simulated by eight zero-length spring elements, of which the tension strength is ignored. Each SMA washer set was modeled by a zero-length spring element, of which the assigned material was the *Self-centering material model*. The FE model is shown in Figure 10. It is worth noting that the nonlinear material properties of the normal concrete and the reinforcement can be considered by the *Concrete02 material model* and *Steel02 material model*, respectively. The geometric nonlinearity of the RC pier can also be accounted for by the co-rotational geometric transformations [39]. The numerical simulation results and the test ones are presented in Figure 11, which shows that the analysis results match well with the test regarding the amplitude of the lateral load and the trend of the hysteretic curve. It confirms that the numerical simulation method proposed in this study can be used to capture the overall rocking behavior of the rocking bridge system.

## 5. Seismic Responses of the Resilient Bridges with SMA-Based Rocking Piers

In order to demonstrate the superiorities of the proposed bridge system in resilience enhancement, four different bridge systems are studied. The first bridge system is a conventional bridge with a RC fixed pier. The second one is a conventional bridge with an ECC-reinforced fixed pier. The third one is a resilient bridge with a RC rocking pier with SMA washers. The last one is a resilient bridge with an ECC-reinforced rocking pier with SMA washers.

### 5.1. FE Model of Resilient Bridge Systems

A comprehensive FE model of the resilient continuous RC box girder bridge is established in OpenSees, as demonstrated in Figure 12. The RC girder (i.e., 20 + 20 m) of the bridge is modeled by 35 elastic beam-column elements. The rocking RC or ECC-reinforced pier is 8.0 m high which is modeled by six displacement-based beam-column fiber elements. The *Concrete02 material model* is used to model the properties of the normal concrete, of which the tension strength is considered. The *ECC02 material model* is employed to simulate the ECC material. The *Steel02 material model* is used for modeling the reinforcing steel. The heights of the two separated pile caps are both 2.0 m, which are modeled using rigid elements. The width of each expansion joint is 0.25 m, which is modeled by a zero-length element. Each bent cap and abutment are modeled by two rigid beam elements along the transverse direction of the bridge. Similarly, the transverse beams in the girder are simulated by two rigid beam elements. Two conventional sliding bearings are placed on each abutment and two fixed bearings are installed on the bent cap above the middle pier. The frictional coefficient of the sliding bearing is 0.02. Each bearing is simulated by a zero-length element connecting the girder and the bent caps or the abutments. The soil and structural interaction of each pile foundation has been considered by six zero-length elements [49,50]. The compressive strength of the conventional concrete for the box girder is 50.0 MPa. The maximum compressive strength and the crush strain of the ECC material in the pier and the rocking pile caps are 80.0 MPa and 1.2%, respectively. The maximum compressive strength and the crush strain of the conventional concrete material in the pier and the rocking pile caps are 40.0 MPa and 1.0%, respectively. The diameters of the longitudinal reinforcement and stirrup used in the pier are 32 mm and 16 mm, of which the corresponding yielding strengths are 440.0 MPa and 300.0 MPa, respectively. A total of 72 reinforcing steels are uniformly arranged around the perimeter of the ECC-reinforced pier, resulting in a reinforcement ratio of 2.28%. The net thickness of the cover concrete or ECC material is 0.05 m. The stirrup interval is 0.1 m at the plastic hinge region and 0.15 m at elsewhere. The SMA washer set is modeled by a compression-only element with a *self-centering material model* and *an Elastic-perfectly plastic material model* in parallel. The *self-centering material model* is used to model the superelasticity of the SMA washer set and the *elastic-perfectly plastic material model* is employed to simulate the precompression by designated an appropriate initial deformation. For instance, the maximum compressive deformation of each SMA washer set is 0.06 m, half of which is consumed to impose prestressed force on the rocking pier.

Two additional conventional bridge systems and a resilient bridge system with a RC rocking pier are also considered to demonstrate the super resilience of the innovative bridge with an ECC-reinforced rocking pier. The main difference between the conventional bridge and the resilient bridge is that the 10 m high fixed pier in the conventional bridge is replaced by an 8 m high pier and a 2 m rocking upper pile cap in height. The FE models of the other three bridge systems can be easily adjusted from that shown in Figure 12 and are therefore not elaborated.

### 5.2. Earthquake Ground Motions

A suite of ground motions being compatible to the acceleration spectra (i.e., E1 and E2 levels) [51,52] are generated and each suite includes seven earthquake ground motions, as shown in Figure 13a,b, respectively. The exceedance probabilities of E1 and E2 level earthquakes in a recurrence interval of 50 years are 10% and 2.5%, respectively. The damping ratio of the acceleration spectra is 5%.

### 5.3. Comparison of Seismic Responses between the Conventional and Resilient Bridges

To evaluate the damage state of the RC and ECC-reinforced pier after earthquake excitation, sectional characteristic analyses are conducted prior to analysis of bridge system. The equivalent bending moments versus curvatures of the sections are shown in Figure 14, where the equivalent bending moments of the RC and ECC-reinforced sections are 19,230.0 kNm and 28,023.0 kNm, respectively, and the associated curvatures are 0.003 1/m and 0.0044 1/m, respectively. The figure indicates that the elastic stiffness and the yield strength of the ECC-reinforced section are both much larger than those of the RC section.

The curvature ductility and the drift ratio of the pier as well as the shear deformation of the bearing are usually selected as the damage indicators for seismic performance assessment of bridges with rocking piers. Therefore, these indicators are used to make a comparison between the conventional and the resilient bridge systems under the considered earthquakes. The curvature ductility indicates the damage state of the plastic hinge region of the pier during earthquakes. Drift ratio is employed to assess the lateral deformation of the bridge system. The uplift ratio indicates the rocking amplitude of the rocking pier, which is expressed as *c/*(*a + b*) shown in Figure 15, where *c* is the maximum uplift distance of the pier; *a* is the distance between the outmost edge of the upper pile cap and the left side of the stress block; *b* is the width of the compressive stress block (i.e., rocking zone). The maximum seismic responses regarding the aforementioned damage indicators of the conventional (fixed base) and the resilient (rocking), bridges with the normal RC piers under E1 and E2 earthquakes are summarized in Table 1 and Table 2, respectively.

The results summarized in Table 1 reveal that the average maximum values of the curvature ductility of the pier, bearing deformation, the drift ratio, and the residual drift ratio of the conventional RC bridge are all larger than those of the resilient RC bridge under E1 level earthquakes. The curvature ductility responses of the RC piers of the conventional bridge and the resilient bridge are both less than 1.0, which means that the RC piers in two bridge systems keep linear state under E1 level earthquakes. The drift ratio of the resilient bridge is 0.88%, which satisfied the principle of the seismic design objective. All the seismic responses confirm that the two bridge systems are both safe under E1 level earthquakes. The average maximum uplift ratio is 0.11%. Even though the earthquake intensity (i.e., E1 level) is not large, the unique property, such as rocking, of the resilient bridge is well exhibited.

When the earthquake intensity increases from E1 to E2 level, the average maximum curvature ductility of the conventional RC bridge sharply increases from 0.94 to 2.99. It implies that the RC pier experiences severe damage when the bridge is subjected to E2 level earthquakes although it performs linearly under E1 level earthquakes. A similar trend also can be found from the seismic performance of the resilient RC bridge system even though the increasing amplitude is not so large compared with the conventional bridge. To gain a thorough understanding about the rocking mechanism of the pier, the typical shear force against drift ratio of the bridge, and the bending moment versus curvature ductility of the section at the bottom of the RC pier under a typical ground motion (i.e., Earthquake No. 1 at E2 level) are shown in Figure 16 and Figure 17, respectively. From these two figures, it can be concluded that the drift ratio of the conventional RC bridge (i.e., 1.71) is slightly larger than that of the resilient RC bridge (i.e., 1.63), but the maximum curvature ductility of the RC pier and the residual drift ratio of the conventional bridge (i.e., 3.24 and 0.039%) are much larger than those of the resilient bridge (i.e., 1.47 and 0.009%). It confirms that the resilient rocking bridge system can significantly alleviate the seismic damage to the RC pier because the dominate period of the rocking bridge is elongated. The seismic response of a SMA washer set is displayed in Figure 18, where the compressive deformation is 0.021 m that is less the uplift threshold of the rocking pier (i.e., 0.03 m). The hysteretic loop response as shown in Figure 18 demonstrates that the SMA washer sets dissipates a lot of energy during earthquakes.

The maximum and average maximum seismic responses regarding the aforementioned damage indicators of the conventional and the resilient bridges with the ECC-reinforced piers under E1 and E2 level earthquakes are summarized in Table 3 and Table 4, respectively.

The results presented in Table 3 show that all the average maximum responses of the resilient bridge except for the curvature ductility of the ECC-reinforced pier are smaller than those of the conventional bridge under E1 earthquakes. Even though the maximum average curvature ductility of the resilient bridge is larger than that of the conventional one, it is still less than 1.0. In other words, the ECC-reinforced pier in the resilient bridge always experiences elastic state under E1 earthquakes.

When the earthquake intensity increased from E1 to E2 level, all the average maximum responses sharply enhanced except the curvature ductility of the ECC-reinforced pier of the resilient bridge increases slightly (i.e., from 0.74 to 0.98). The average maximum responses of the bearing deformation of the conventional and the resilient ECC-reinforced bridges are similar under E2 level earthquakes, but the average maximum curvature ductility of the ECC-reinforced pier and the residual drift ratio of the resilient bridge (i.e., 0.98 and 0.011%) are much smaller than those of the conventional bridge (i.e., 1.80 and 0.073%). The ECC-reinforced pier in the resilient bridge system can always keep in linear performance under E2 earthquakes but it may yield in the conventional bridge system under the same level earthquakes. The average maximum drift ratio of the conventional bridge is larger in comparison with the resilient bridge because the ECC-reinforced pier in the conventional bridge yields but the ECC-reinforced pier in resilient bridge keeps a linear behavior. To take a close look at the seismic response, the typical time history responses of the lateral seismic force versus drift ratio of the ECC-reinforced conventional and the ECC-reinforced resilient bridges under a typical earthquake (i.e., Earthquake No.1 at E2 level) are illustrated in Figure 19. It can be confirmed that although the two bridges experienced similar lateral displacement, the rocking effect contributes to the most part of the total lateral displacement but the fixed pier mainly depended on the flexural deformation of the pier itself. The seismic responses of the section at the bottom of the plastic hinge region in two bridge systems are presented in Figure 20, which implies that the fixed base ECC-reinforced pier suffered severe damage but the rocking pier stayed elastic. The seismic response of one SMA washer set is displayed in Figure 21, where the self-locking effect is triggered when the compressive deformation reached 0.03 m. Once the maximum drift ratio of the resilient bridge reaches 2.0%, the self-locking effect will act and the ECC-reinforced pier will yield simultaneously. In other words, if the drift ratio of the resilient bridge exceeds 2.0%, the incremental lateral displacement will completely depend on the yielding deformation of the ECC-reinforced pier.

### 5.4. Comparison of Seismic Responses between the RC and ECC-Reinforced Resilient Bridges

The aforementioned comparisons demonstrates that the ECC-reinforced bridge systems are more resilient than the conventional bridge systems. The following part will make a comparison between the two resilient bridge systems with the RC rocking pier and the ECC-reinforced rocking pier. The average maximum drift ratio of the resilient bridge with RC rocking bridge under E2 level earthquakes is 1.49 but the value of the ECC-reinforced resilient rocking bridge is 1.70. The reason is that the yielding strength of the RC pier is smaller than the ECC-reinforced pier so that it cannot sustain large rocking amplitude. It can reconfirm by the response of the average maximum curvature ductility of two bridges. For instance, the average maximum curvature ductility of the resilient bridge with RC rocking bridge under E2 level earthquake is 1.50, whereas the value of the ECC-reinforced resilient rocking bridge is 0.98. A case is selected for further investigation: the drift ratio versus lateral seismic force responses of two resilient bridges subjected to a typical earthquake (i.e., Earthquake No. 1 at E2 level) is shown in Figure 22. From Figure 23, it can be recognized that the maximum drift ratio of the resilient bridge with the ECC-reinforced pier is 2.07, but the corresponding value is only 1.63. The curvature ductility versus bending moment responses of two resilient bridges under a typical earthquake (i.e., Earthquake No. 1 at E2 level) is shown in Figure 23. The maximum curvature ductility of the RC pier is 1.47 but the counterpart of the ECC-reinforced pier is 1.07. The earthquake-induced damage of the ECC-reinforced pier is so small (i.e., 1.07), which needs no repair but the RC pier should be retrofitted after earthquake.

## 6. Conclusions

This study presented a resilient self-centering bridge system with the ECC-reinforced rocking pier. An experimental study on the ECC-reinforced column was employed to demonstrate the accuracy of the proposed constitutive model of the ECC material implemented in OpenSees. The advantages of the resilient ECC-reinforced rocking bridge system, such as low damage and negligible residual deformation, were verified by conducting a series of nonlinear seismic analyses on both resilient and conventional bridges with RC or ECC-reinforced piers. Several comments and conclusions are summarized as follows:All the conventional and the resilient rocking bridge systems with the RC pier or the ECC-reinforced pier can satisfy the seismic design objectives under E1 level earthquakes;The proposed resilient rocking bridge system with the RC pier or the ECC-reinforced pier shows superior seismic performance over the conventional bridge in terms of the response such as the curvature ductility of the pier, bearing deformation, drift ratio, and residual drift ratio of the bridge under E2 level earthquakes;The resilient rocking bridge with the ECC-reinforced pier can achieve superior damage retrofitting capacity than the resilient rocking bridge with RC pier. The average maximum curvature ductility of the ECC-reinforced pier was only 0.98 but the counterpart of the RC pier was 1.5 which indicates the RC experienced yielding state during E2 level earthquakes;Due to the protection mechanism (i.e., self-locking effect) of the SMA washer spring device against overload and the super resilient property of the ECC material, the resilient rocking bridge system shows the excellent damage control capacity;The SMA washer spring device cannot only provide restoring force for the resilient bridge system but also can dissipate moderate earthquake energy input.

## Figures and Tables

**Figure 1 materials-14-06500-f001:**
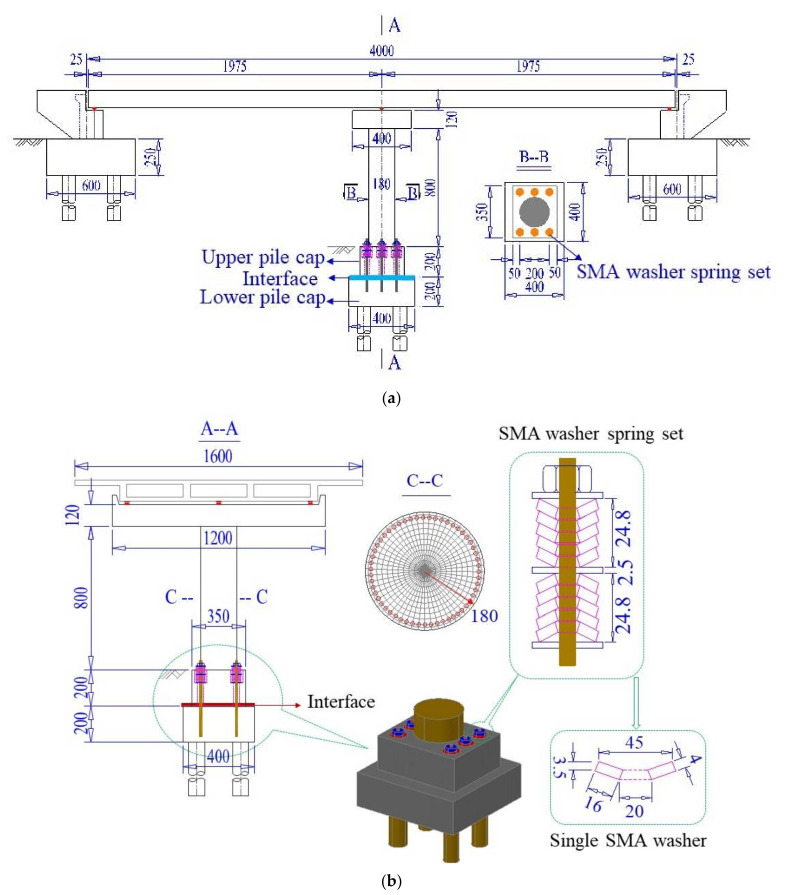
Configuration of the resilient bridge with ECC-reinforced rocking pier (Unit: cm). (**a**) Elevation; (**b**) Layout diagram of the SMA washer spring devices.

**Figure 2 materials-14-06500-f002:**
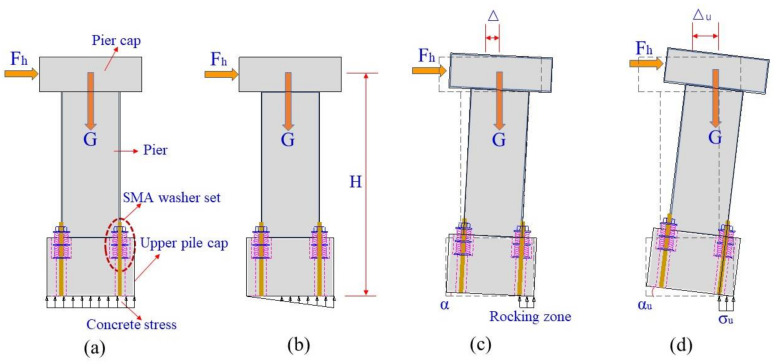
Rocking mechanism of the SMA washer-based pier. (**a**) Original state; (**b**) Onset of uplift; (**c**) Rocking state; (**d**) Self-locking state.

**Figure 3 materials-14-06500-f003:**
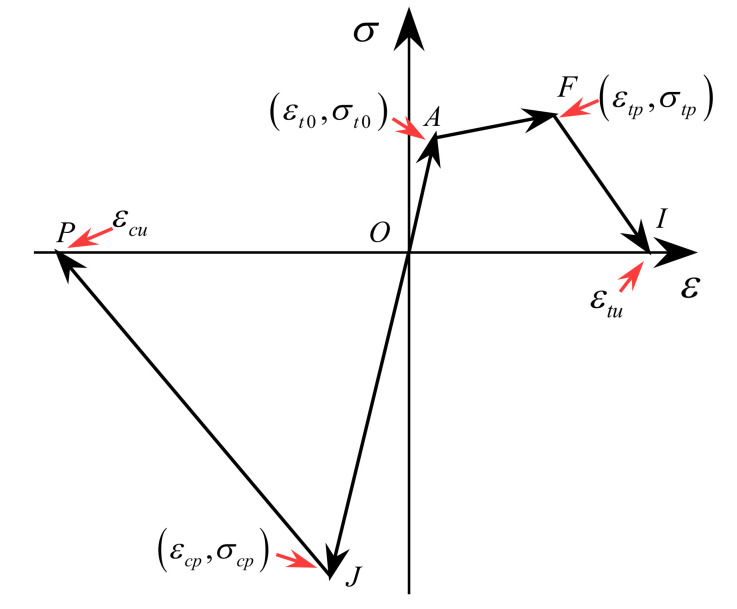
Envelop curves of the ECC constitutive model [46].

**Figure 4 materials-14-06500-f004:**
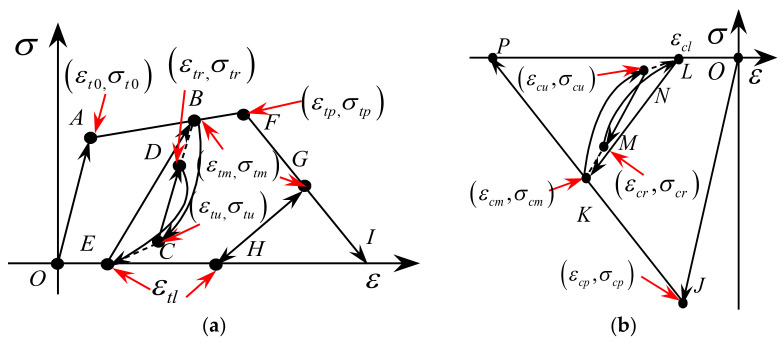
Unloading and reloading rules of the ECC model (**a**) Tension region; (**b**) Compression region [46].

**Figure 5 materials-14-06500-f005:**
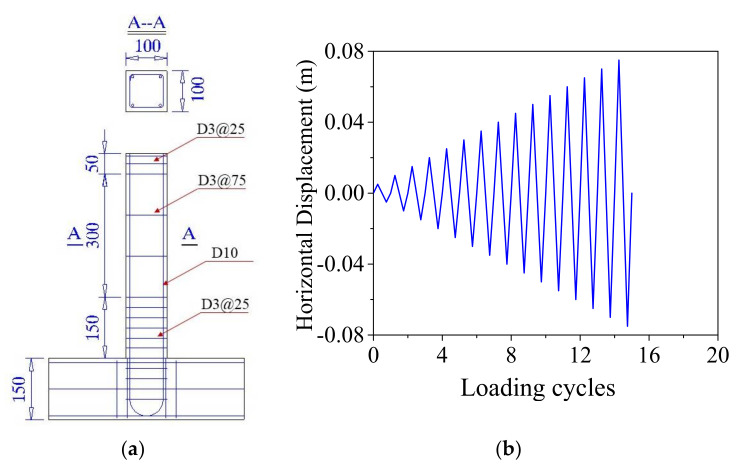
Specimen and loading protocol. (**a**) ECC-reinforced column specimen (Unit: mm); (**b**) Loading protocol for quasi-static test.

**Figure 6 materials-14-06500-f006:**
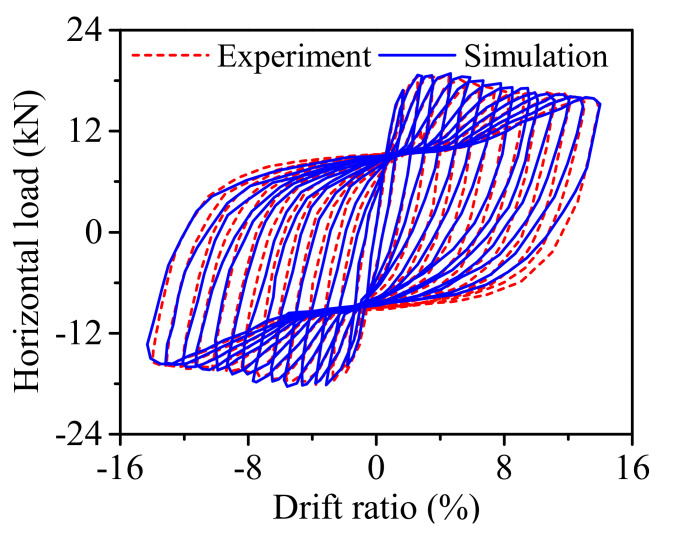
Numerical simulation results against test results.

**Figure 7 materials-14-06500-f007:**
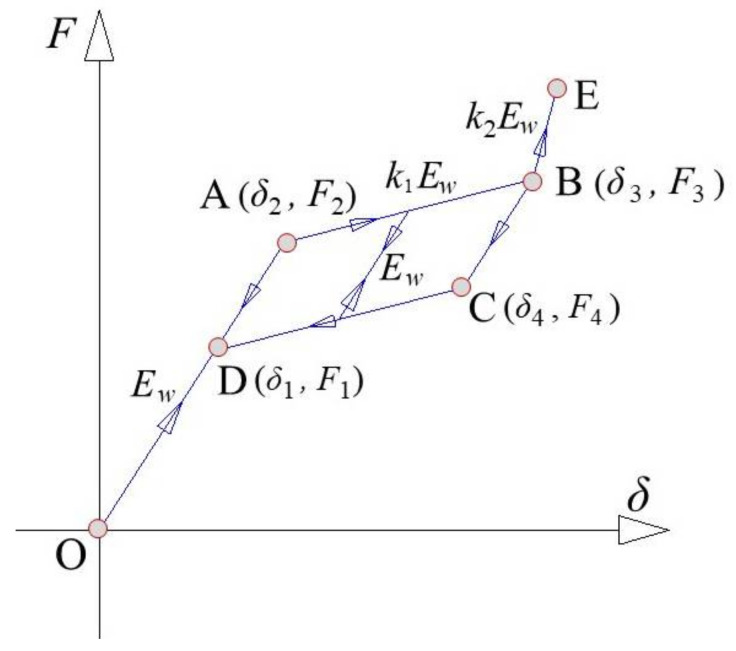
Idealized constitutive model for SMA washer.

**Figure 8 materials-14-06500-f008:**
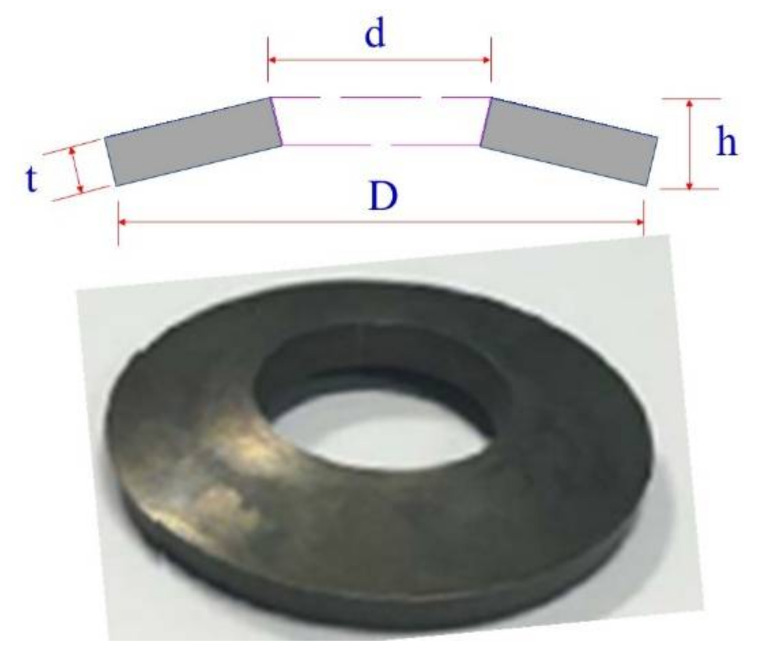
Configuration of a typical SMA washer.

**Figure 9 materials-14-06500-f009:**
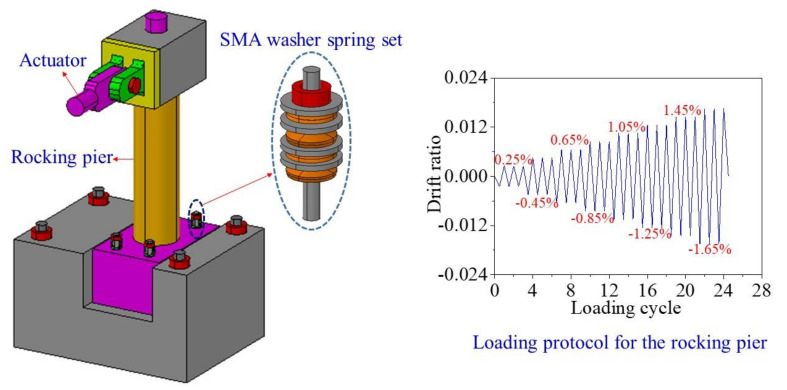
Configuration of a rocking pier specimen and loading protocol.

**Figure 10 materials-14-06500-f010:**
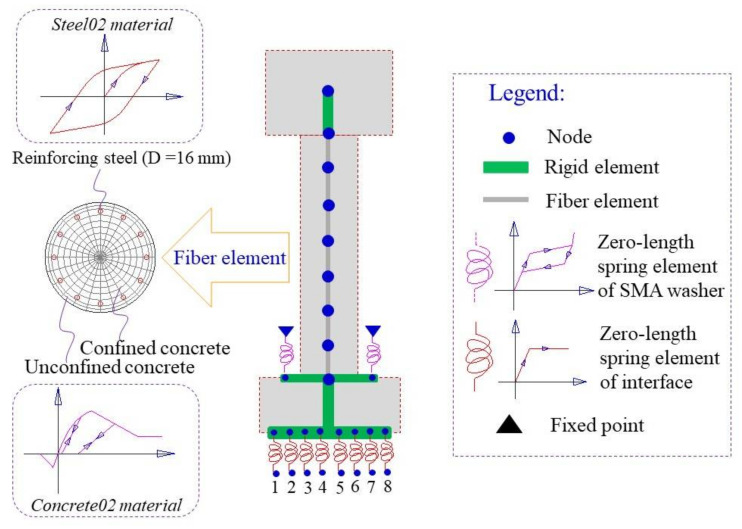
FE model of the rocking pier specimen.

**Figure 11 materials-14-06500-f011:**
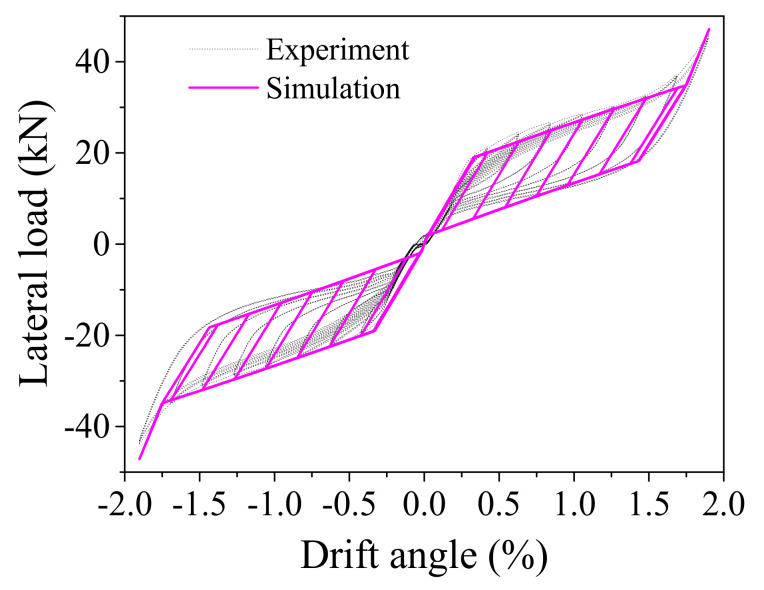
Comparison between numerical simulation and test results.

**Figure 12 materials-14-06500-f012:**
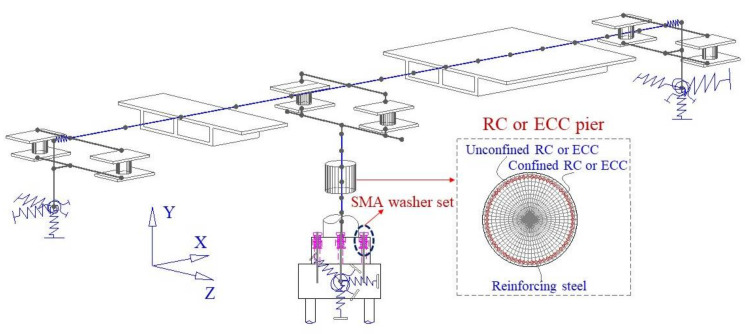
FE model of the resilient bridge with SMA-based rocking pier.

**Figure 13 materials-14-06500-f013:**
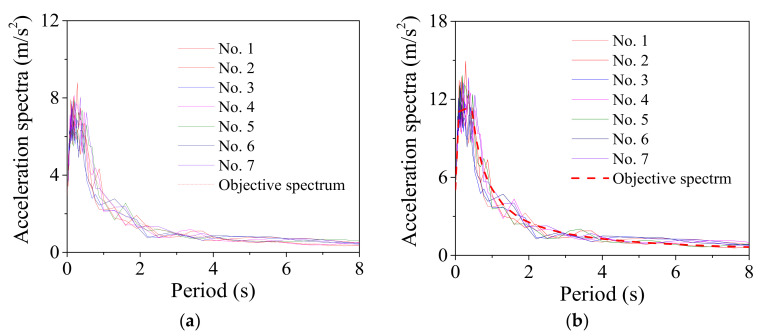
Earthquake spectra together with earthquake motions. (**a**) E1 level; (**b**) E2 level.

**Figure 14 materials-14-06500-f014:**
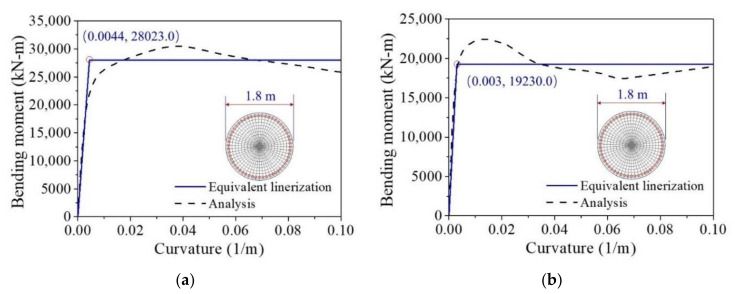
Sectional analyses of bending moment vs. curvature. (**a**) RC; (**b**) ECC.

**Figure 15 materials-14-06500-f015:**
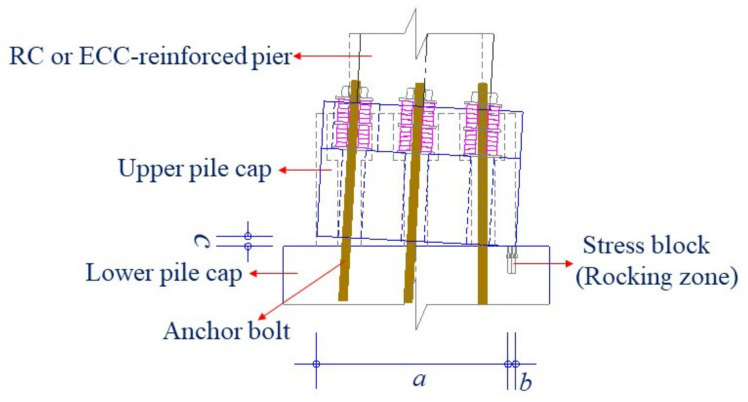
Schematic diagram of the uplift ratio.

**Figure 16 materials-14-06500-f016:**
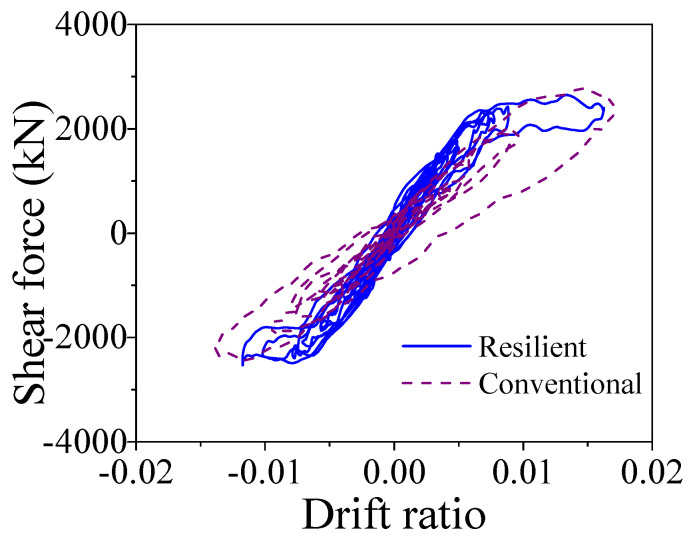
Shear force vs. drift ratio.

**Figure 17 materials-14-06500-f017:**
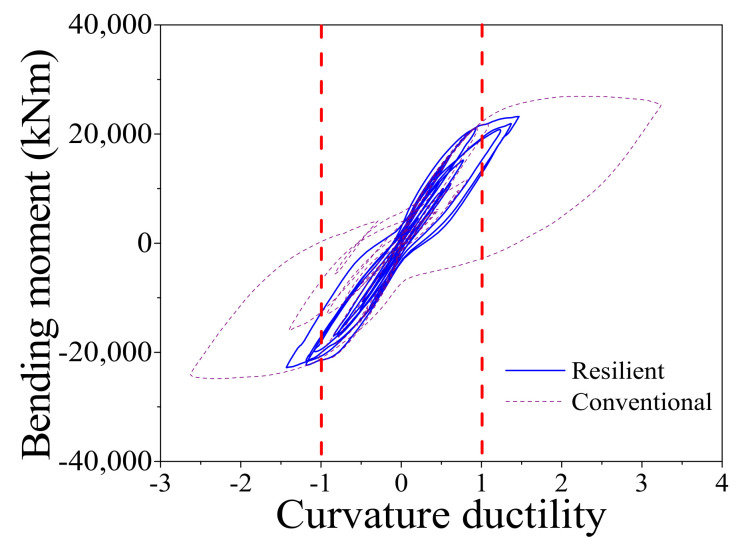
Bending moment vs. curvature ductility.

**Figure 18 materials-14-06500-f018:**
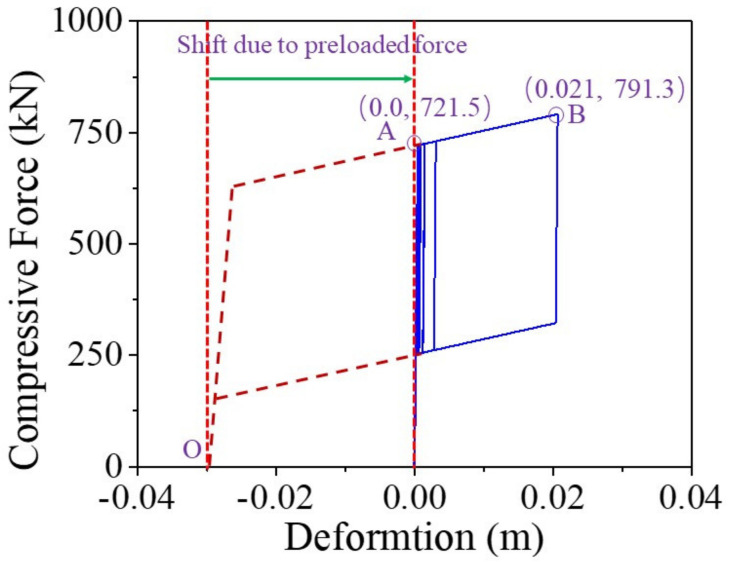
Compressive force vs. deformation of SMA washer set.

**Figure 19 materials-14-06500-f019:**
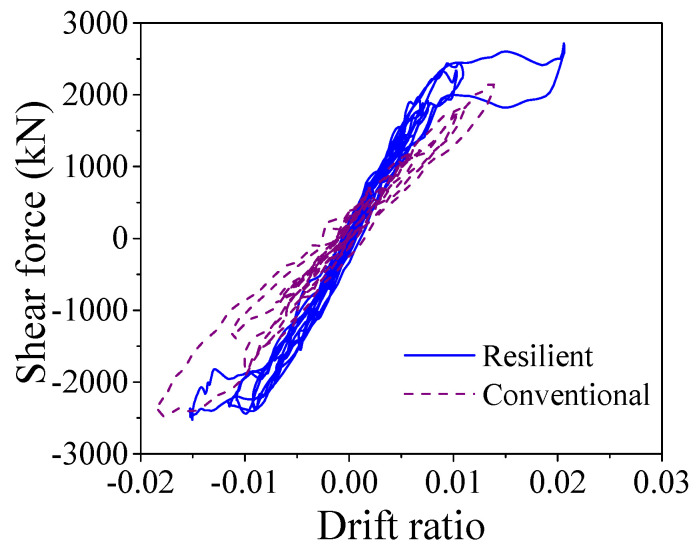
Shear force vs. drift ratio.

**Figure 20 materials-14-06500-f020:**
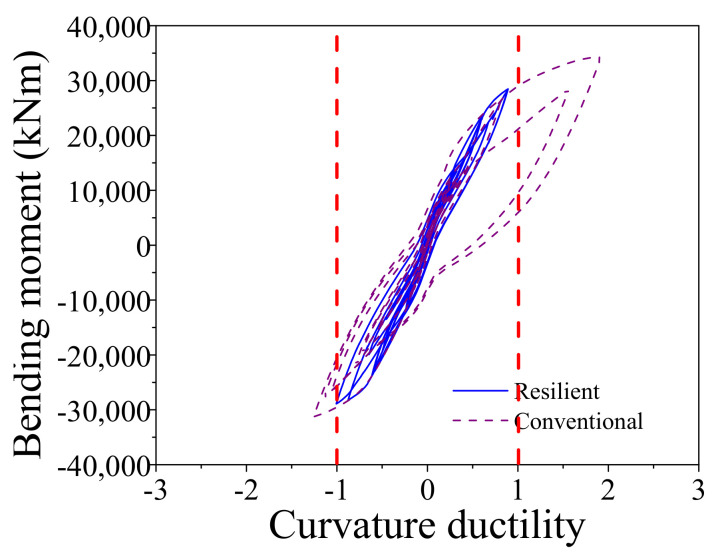
Bending moment vs. curvature ductility.

**Figure 21 materials-14-06500-f021:**
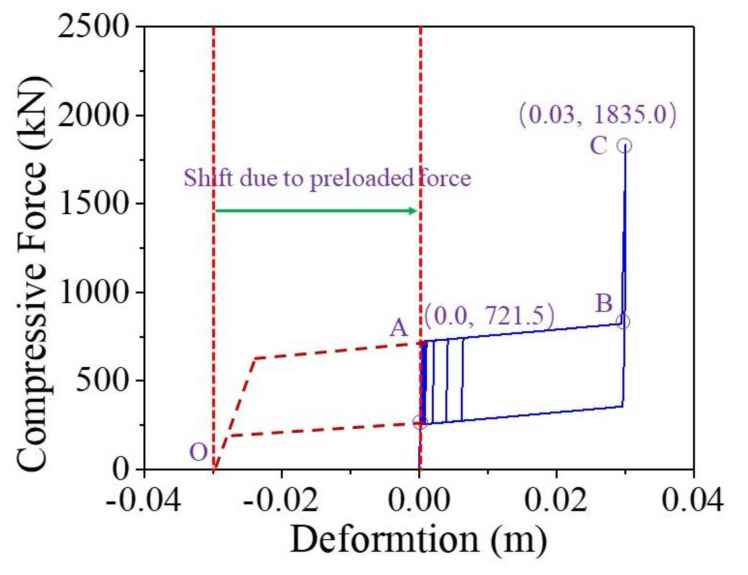
Compressive force vs. deformation of SMA washer set.

**Figure 22 materials-14-06500-f022:**
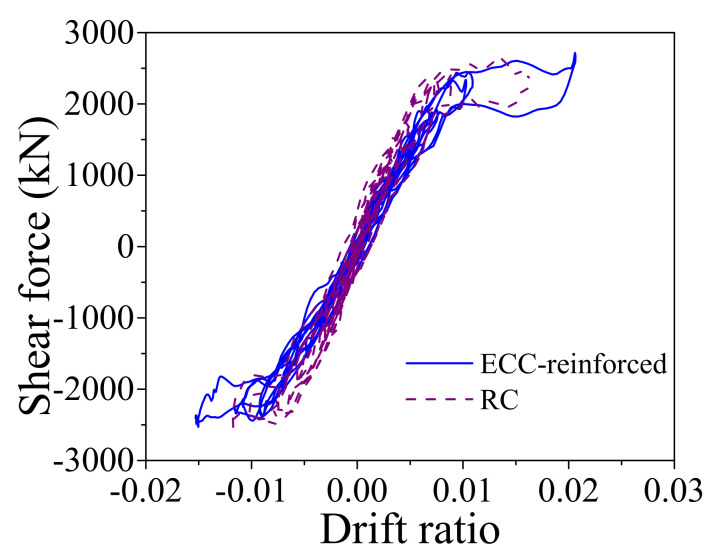
Shear force vs. drift ratio.

**Figure 23 materials-14-06500-f023:**
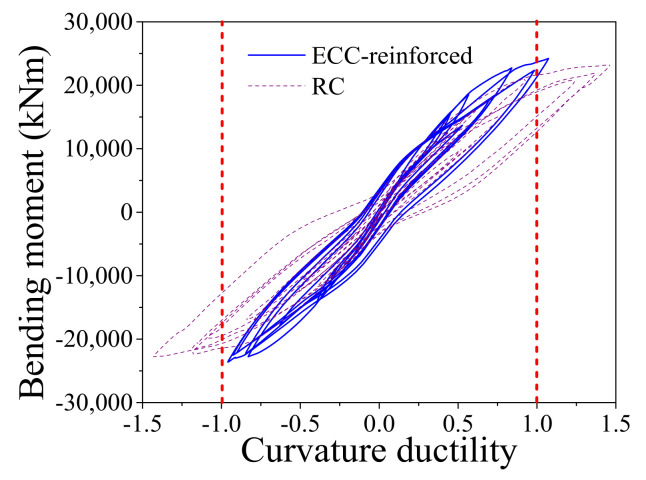
Bending moment vs. curvature ductility.

**Table 1 materials-14-06500-t001:** Maximum seismic responses of RC bridges at E1 level.

Earthquake	Curvature Ductility		Bearing Deformation (cm)		Drift Ratio (Residual Drift Ratio) (%)		Uplift Ratio (%)
No.	Conventional	Resilient	Conventional	Resilient	Conventional	Resilient	Resilient
1	1.15	0.76	11.99	7.34	1.01 (0.005)	0.62 (0.008)	0.07
2	1.15	0.92	11.91	8.70	1.00 (0.029)	0.72 (0.008)	0.08
3	0.80	0.82	9.30	7.68	0.79 (0.002)	0.65 (0.001)	0.07
4	0.89	1.17	10.20	11.75	0.86 (0.011)	0.99 (0.013)	0.30
5	0.74	0.70	8.77	6.95	0.74 (0.014)	0.57 (0.009)	0.06
6	0.83	1.09	9.86	9.80	0.81 (0.027)	0.83 (0.006)	0.17
7	0.99	0.68	10.81	6.54	0.91 (0.002)	0.56 (0.008)	0.05
Avg. value	0.94	0.88	10.41	8.40	0.88 (0.013)	0.70 (0.008)	0.11

**Table 2 materials-14-06500-t002:** Maximum seismic responses of RC bridges at E2 level.

Earthquake	Curvature Ductility		Bearing Deformation (cm)		Drift Ratio (Residual Drift Ratio) (%)		Uplift Ratio (%)
No.	Conventional	Resilient	Conventional	Resilient	Conventional	Resilient	Resilient
1	3.24	1.47	20.33	19.43	1.71 (0.039)	1.63 (0.009)	1.01
2	2.70	1.73	18.38	20.77	1.54 (0.022)	1.74 (0.002)	1.12
3	3.00	1.47	18.96	16.42	1.59 (0.057)	1.38 (0.020)	0.70
4	3.23	1.56	19.02	17.58	1.58 (0.054)	1.47 (0.009)	0.86
5	2.95	1.31	18.37	16.17	1.54 (0.060)	1.36 (0.005)	0.73
6	2.68	1.47	18.97	16.68	1.57 (0.046)	1.39 (0.011)	0.72
7	3.15	1.49	19.52	17.89	1.64 (0.040)	1.50 (0.001)	0.86
Avg. value	2.99	1.50	19.08	17.85	1.60 (0.045)	1.49 (0.008)	0.86

**Table 3 materials-14-06500-t003:** Maximum seismic responses of ECC-reinforced bridges at E1 level.

Earthquake	Curvature Ductility		Bearing Deformation (cm)		Drift Ratio (Residual Drift Ratio) (%)		Uplift Ratio (%)
No.	Conventional	Resilient	Conventional	Resilient	Conventional	Resilient	Resilient
1	0.67	0.76	11.93	10.23	0.98 (0.013)	0.87 (0.014)	0.08
2	0.62	0.93	11.17	15.03	0.94 (0.026)	1.26 (0.005)	0.49
3	0.68	0.57	11.30	8.35	1.00 (0.035)	0.69 (0.016)	0.06
4	0.65	0.87	11.74	13.63	0.97 (0.021)	1.10 (0.017)	0.32
5	0.63	0.62	11.13	8.66	0.94 (0.011)	0.73 (0.011)	0.06
6	0.71	0.72	12.60	10.06	1.04 (0.035)	0.83 (0.018)	0.08
7	0.61	0.74	10.91	10.18	0.92 (0.003)	0.86 (0.014)	0.08
Avg. value	0.65	0.74	11.54	10.88	0.97 (0.021)	0.91 (0.014)	0.17

**Table 4 materials-14-06500-t004:** Maximum seismic responses of ECC-reinforced bridges at E2 level.

Earthquake	Curvature Ductility		Bearing Deformation (cm)		Drift ratio (Residual Drift Ratio) (%)		Uplift Ratio (%)
No.	Conventional	Resilient	Conventional	Resilient	Conventional	Resilient	Resilient
1	1.84	1.07	22.17	24.66	1.84 (0.137)	2.07 (0.007)	1.48
2	1.66	0.99	21.26	20.89	1.78 (0.040)	1.75 (0.006)	1.08
3	1.88	0.93	22.15	19.31	1.86 (0.065)	1.62 (0.006)	0.93
4	2.21	0.98	24.30	18.45	2.02 (0.080)	1.32 (0.029)	0.83
5	1.51	0.99	19.90	19.96	1.67 (0.131)	1.67 (0.011)	0.98
6	2.04	0.92	23.24	18.54	1.93 (0.022)	1.54 (0.011)	0.84
7	1.47	0.98	19.69	23.32	1.66 (0.037)	1.95 (0.009)	1.37
Avg. value	1.80	0.98	21.81	20.73	1.82 (0.073)	1.70 (0.011)	1.07

## Data Availability

The data presented in this study are available on request from the corresponding author.

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
