# Peer review of "Seismic Response of Resilient Bridges with SMA-Based Rocking ECC-Reinforced Piers"

_materials, 2021, doi:10.3390/ma14216500_

Round 1
Reviewer 1 Report
The manuscript investigated the seismic response of Novel Bridges with SMA-based Rocking ECC-reinforced Piers. The manuscript has some serious technical flaws for which the reviewer does not recommend accepting the paper. Based on the review of the manuscript, the reviewer has the following comments:
- The concept of SMA-washer based rocking pier is nothing new as it has been extensively studied experimentally in the following article:
Rocking bridge piers equipped with shape memory alloy (SMA) washer springs. Engineering Structures; Volume 214, 1 July 2020, 110651
The only new thing the authors considered in this study is ECC in the bottom of the pier. Also the SMA washer arrangement shown in different figures raised significant concerns. For example, Fig. 1 shows that the SMA-washer device is fully embedded inside the upper pile cap. However, in Fig. 2 shows the SMA-washer device outside the upper pile cap.
- Which composition of SMA has been used as the SMA washer? Ni-based, Fe-based or Cu-based? Nothing is mentioned about the SMA properties. Fig. 6 shows the idealized behavior of SMA washer but no information is provided on the value of each parameter shown in this Figure. This information is very crucial for this study. It is not acceptable just to show a generalized hysteretic response of SMA material.
- More details are required for the Opensees model adopted here. What are the constitutive models used for SMA, ECC, reinforcing steel, normal concrete and so on?
- Proper validation of the adopted numerical model must be provided. The authors did validate the material model for ECC, however, it is not enough to capture the overall rocking behavior of the column as well as the full bridge. The authors should validate their numerical models with the experimental results of the study mentioned in comment-1.
- It is not clear how the precompression was applied in the numerical model? Please explain.
- Section 2.2: The authors mentioned that “The natural period of the novel bridge is much larger than that of the conventional bridge, which makes it far away from the dominant periods of the earthquake ground motions.” When two bridges are far away in-terms of their fundamental period, how it is justified to compare the seismic response of these two structures. These two structures are not dynamically comparable since their dynamic behavior is significantly different.
- 2 shows the rocking response of a cantilever column. However, in the FE model the authors considered fixed bearing at the pier location? These two scenarios cannot be the same. Please justify.
- Although the authors started the paper highlighting the importance of controlling residual drift, however, no results are presented for residual drift of the so-called novel bridge pier.
Reviewer 2 Report
In general, the authors spend a lot of time to draft this manuscript.
It contains some interesting materials and results.
My main concern is about the novelty:
First of all, the "title" need to be revised. The term "novel bridge" does not sound right to me.
Second, the authors should clearly "itemize" all the novelties in this manuscript compared to the published ones and distinguish what is completely new and what aspects are only improved compared to the previous publications.
Third, I feel that there is a bias in referencing. 27 out of 37 references are from Chinese authors. It is hard to accept that 70%+ of world research in this field is done in China. This is a critical concern and I am already aware some very relevant works that are not mentioned (which is indeed questions the novelty of the current manuscript). This should be revolved.
Author Response
Point 1: First of all, the "title" need to be revised. The term "novel bridge" does not sound right to me.
Response 1:Thanks for the comment. We agree, the title has been changed to “Seismic Response of Resilient Bridges with SMA-based Rocking ECC-reinforced Piers”
Point 2: Second, the authors should clearly "itemize" all the novelties in this manuscript compared to the published ones and distinguish what is completely new and what aspects are only improved compared to the previous publications.
Response 2: Thanks for the comment. The novelties have been clearly itemized though the manuscript.
Point 3: Third, I feel that there is a bias in referencing. 27 out of 37 references are from Chinese authors. It is hard to accept that 70%+ of world research in this field is done in China. This is a critical concern and I am already aware some very relevant works that are not mentioned (which is indeed questions the novelty of the current manuscript). This should be revolved.
Response 3:The comment is gratefully acknowledged. Another 11 references have been appended to the manuscript (see pages 21~22).
Reviewer 3 Report
Paper deals with presentation of the advantages of the rocking pier solution made from ECC and connected to the base with application of SMA washers. It is not clear when reading the paper how shape memory alloy is activated (In general SMA materials are activated with the temperature). After reading the paper one can conclude that these washers are simply working in their elastic range when pier is rocking, causing dissipation of energy through friction. The reason why SMA concept is used here needs better explanation.
The concept of the paper is rather clear: Authors are presenting comparison of numerical solutions (with application of OpenSees software) obtained for two different solutions (one with ECC material and the second with RC material). In Fig. 5 Reader may find two graphs which may be treated as some kind of numerical model validation. Of course, this comparison gives us only some general estimation of the model’s validity. In case of experimental data, the tangent of the graph for small deformation is smaller than in case of numerical model (in numerical model it is almost vertical line). The only conclusion from this comparison is that the graph boundaries on vertical and horizontal axes are similar, but the relationship between force and drift ratio is rather not the same. Explaining this validation is crucial because all conclusions from the paper are based on numerical calculation.
When we assume that this validation in case of ECC is correct, then what about RC material? We can’t find any RC validation, but presentation of the results and conclusions are based on comparison of solutions with ECC and RC.
Going further, constitutive models used in the paper are not described properly. We have some concepts (like presented in Fig.3 or Fig.6) without any equation or parameter, and then somewhere in the text we have material data which can’t be connected with these explanations or symbols given in the drawings. The constitutive models should be explained in detail with proper equations and material parameters. Without clearance in that part paper can’t be accepted for publication (Authors should give to the Reader all data needed to repeat the analysis if needed).
In my opinion there is no proper explanation of numerical model used. In general, the whole bridge structure is modeled with beam elements (see Fig. 8). I am not sure if such model is really physical in case of such complicated structure like the bridge. There is no information about number of elements used (maybe the solution would be different for different number of elements). Please work on this issue.
The Conclusions part is too general. It would be good to give there more details about obtained results. In my opinion when comparing Fig.12 with Fig.15 and Fig. 13 with Fig.16 no clear conclusions about the advantages of one solution over the other can be drawn.
Detailed remarks:
- Keywords: “analysis” is not a keyword;
- What is the meaning of zigzags on the right and left of Fig. 1a?
- Fig.2d – “u” should be the lower index for “delta”.
- Lines 165-166 – once strains without % and once with them – make it consistent through the paper.
- Line 168: “ultra-strength”? Is it a notion from strength of materials? Maybe “ultimate uniaxial strength”?
- Fig. 5a – too low quality of the graph.
- The unit of bending moment is rather kNm not kN-m.
- Fig. 8 – could You possibly explain the mining of the elements connecting the transverse elements.
- Fig. 8 - What is the meaning of element plotted with the symbol like capacitor in electricity?
- Fig.14. Quite visible differences in font size between Fig.13 and Fig. 14 (Fig. 16 and Fig. 17).
- Line 387 – “mode” – maybe “model”?
Round 2
Reviewer 1 Report
The authors have addressed all comments satisfactorily. The paper now can be accepted for publication.
Author Response
The comments from the reviewer are gratefully acknowledged.
Reviewer 2 Report
Although I provided only 3 comments I do not see they are fully addressed.
I do not see the itemized novelty section.
I still see some novel bridge inside the paper.
I do not see any cut in the references.
Author Response
Added references:
- Priestley MJN, Tao J. Seismic response of precast prestressed concrete frames with partially debonded tendons, PCI J, 38(1):58–69, 1993.
- H Roha, A M. Reinhorn. Hysteretic behavior of precast segmental bridge piers with superelastic shape memory alloy bars, Eng Struct, 32: 3394-3403, 2010.
- H Roh, AM Reinhorn. Analytical modeling of rocking elements, Eng Struct, 31(5): 1179–89, 2009.
- H Roh, AM Reinhorn. Nonlinear static analysis of structures with rocking columns, J Struct Eng, 136(5):532–42, 2009.
Reviewer 3 Report
All my remarks were taken into account in the paper's current form. An article may be published in the Materials journal as it stands.
Author Response

(The authors gave the same response as above.)
